# A Beam Search Framework for Quantum Circuit Mapping

**DOI:** 10.3390/e27030232

**Published:** 2025-02-24

**Authors:** Cheng Qiu, Pengcheng Zhu, Lihua Wei

**Affiliations:** 1School of Computer Science, Nanjing University of Information Science and Technology, Nanjing 210000, China; suolioma@gmail.com; 2College of Information Engineering, Taizhou University, Taizhou 225300, China; angelirene@163.com; 3College of Information Engineering, Suqian University, Suqian 223800, China

**Keywords:** quantum computing, quantum circuit mapping, noisy intermediate-scale quantum computing, limited connectivity, beam search

## Abstract

In the era of noisy intermediate-scale quantum (NISQ) computing, the limited connectivity between qubits is one of the common physical limitations faced by current quantum computing devices. Quantum circuit mapping methods transform quantum circuits into equivalent circuits that satisfy physical connectivity constraints by remapping logical qubits, making them executable. The optimization problem of quantum circuit mapping has NP-hard computational complexity, and existing heuristic mapping algorithms still have significant potential for optimization in terms of the number of quantum gates generated. To reduce the number of SWAP gates inserted during mapping, the solution space of the mapping problem is represented as a tree structure, and the mapping process is equivalent to traversing this tree structure. To effectively and efficiently complete the search process, a beam search framework (BSF) is proposed for solving quantum circuit mapping. By iteratively selecting, expanding, and making decisions, high-quality target circuits are generated. Experimental results show that this method can significantly reduce the number of inserted SWAP gates on medium to large circuits, achieving an average reduction of 44% compared to baseline methods, and is applicable to circuits of various sizes and complexities.

## 1. Introduction

With IBM’s recent launch of the Condor superconducting quantum computer featuring over 1000 qubits [1], Noisy Intermediate-Scale Quantum (NISQ) computing devices [2] have attracted widespread attention in both academia and industry. However, these quantum computing devices still have numerous shortcomings. Their core quantum processing unit (QPU) supports only a limited set of quantum operations and is affected by constraints on connectivity between qubits, as well as the relatively short coherence times of physical qubits, which typically results in a high error rate during the execution of quantum circuits. Current QPUs often cannot directly execute quantum circuits due to limited connectivity between qubits, necessitating the design of quantum circuit mapping methods [3,4] to remap qubits, thereby transforming them into equivalent quantum circuits that fit the architecture of the physical device.

Quantum circuit mapping consists of two parts: the initial mapping of qubits and the remapping. Since the initial mapping $\pi_{0}$ can interfere with the remapping, the latter becomes the primary focus. Existing qubit remapping methods can be classified into two categories. The first category redefines the quantum circuit mapping problem as an optimization problem and uses existing algorithms to solve it [5,6,7,8,9,10,11]. However, the quantum circuit mapping problem on NISQ devices has an NP-complete computational complexity. While these methods can achieve better results for circuits of limited scale, they often lack scalability and practicality when dealing with large-scale circuits. The second category employs heuristic search, progressively transforming the initial circuit into a version suitable for the physical device [2,12,13,14,15,16,17]. Experimental results indicate that heuristic search has advantages over the first category for large circuits and exhibits strong applicability across circuits of various scales but still has shortcomings. Current heuristic search methods struggle to balance search width and depth when addressing the tree-like solution space of the quantum circuit mapping problem, leading to difficulties in maintaining good mapping quality and efficiency for circuits of varying sizes and complexities.

Inspired by the recent significant success of beam search algorithms in natural language text generation [18], this paper proposes a quantum circuit mapping method based on a beam search framework (BSF), aiming to minimize the number of inserted SWAP gates. The main contributions include: (1) the design of a multi-layer window approximation of a real effect value function to calculate the effect value of each inserted SWAP gate node in the search tree, providing a basis for the search decision-making process and (2) the construction of a quantum circuit mapping method based on the aforementioned approximate real effect value function within the beam search framework, subdividing the search process into selection, expansion, and decision phases. This method leverages the characteristics of beam search to adjust the search width and depth, allowing the algorithm to adapt to quantum circuits of different scales.

## 2. Preliminaries

### 2.1. Qubit

Classical computer bits have two states, usually represented as 0 and 1, serving as the fundamental units of information. In contrast, a qubit also has two basic states, typically represented as |0〉 and |1〉. Unlike classical bits, qubits can exist in a linear combination of the two basic states, which can be expressed as |ψ〉=α|0〉+β|1〉, where ${\left|\alpha \right|}^{2}+{\left|\beta \right|}^{2}=1$, and the state vector is (α,β). Additionally, two or more qubits can become entangled, forming a composite system. The state of a two-qubit system can be represented as $\psi =\alpha_{00}|00\rangle +\alpha_{01}|01\rangle +\alpha_{10}|10\rangle +\alpha_{11}|11\rangle$, with its state vector being $\left(\alpha_{00},\alpha_{01},\alpha_{10},\alpha_{11}\right)$.

### 2.2. Quantum Operation

Quantum circuits consist of qubits and a sequence of quantum gates applied to the qubits, collectively representing a quantum program. As shown in Figure 1, each horizontal line in the quantum circuit represents a qubit, while the quantum gates applied to the qubits are represented by different blocks along the lines.

From left to right in Figure 1 are the Hadamard gate, controlled NOT (CNOT) gate, and SWAP gate. The squares represent single-qubit gates, while the vertical lines connecting two qubits represent two-qubit gates. Barenco has demonstrated that any quantum circuit can be represented using only single-qubit gates and CNOT gates [19]. Based on this, the test quantum circuits in the test set contain only single-qubit gates and CNOT gates, and through combinations of these gates, all the quantum gates supported by the IBM cloud computing platform can be constructed.

### 2.3. NISQ Computing Architecture

Figure 2 shows the classical IBM Q Tokyo chip architecture [16], where the effective time of the qubits averages around 50 μs, and the average error probabilities for single-qubit gates, measurement gates, and CNOT gates are denoted as $4.43\times 10^{-3}$, $8.47\times 10^{-2}$, and $3.00\times 10^{-2}$, respectively. As illustrated in Figure 2, two coupled physical qubits are connected by bidirectional arrows. These qubits are arranged in a planar geometric structure, and due to hardware connectivity constraints, a single qubit can only couple with its neighboring qubits. For example, $q_{0}$ is connected to $q_{1}$ and $q_{5}$, meaning that a CNOT gate can be applied to the qubit pair $\left\{q_{0},q_{1}\right\}$ and $\left\{q_{0},q_{5}\right\}$. However, $q_{0}$ is not directly connected to $q_{6}$, so a CNOT gate cannot be directly applied to these two qubits.

### 2.4. Quantum Circuit Mapping

Due to the connectivity constraints between qubits in the quantum physical architecture, quantum programs must undergo quantum circuit mapping before execution to convert logical quantum programs into hardware-compatible quantum circuits. The connectivity constraints primarily affect the execution of two-qubit gates, while they have no impact on the execution of single-qubit gates. Therefore, this study focuses solely on addressing the connectivity constraint issues of two-qubit gates through quantum circuit mapping.

The basic quantum circuit mapping process is as follows: Assuming the quantum physical architecture is as shown in Figure 3, given an original quantum circuit diagram to be processed, as indicated in part b of the figure, only the second CNOT gate can be executed directly. The first CNOT gate cannot be executed immediately because its corresponding logical qubits, $v_{0}$ and $v_{2}$, do not correspond to directly adjacent physical qubits, $q_{0}$ and $q_{2}$, when mapped to the physical architecture. Therefore, a quantum circuit mapping transformation is required.

Current mainstream mapping transformation methods involve inserting SWAP operations into the quantum circuit to exchange the states of the corresponding logical qubits in the physical qubits. This approach allows for adjustments in the mapping state, enabling the execution of two-qubit gates that are originally constrained by connectivity limitations. As shown in Figure 3, after inserting a SWAP gate between the logical qubits $v_{1}$ and $v_{2}$ of the second CNOT gate, all two-qubit gates in the updated front layer satisfy the connectivity constraints. After inserting the SWAP gate, the mapping π updates to $\left\{v_{0}\to q_{0},v_{1}\to q_{2},v_{2}\to q_{1},v_{3}\to q_{3}\right\}$, allowing all quantum gates in the front layer to be executable.

## 3. A Beam Search Framework for Quantum Circuit Mapping

### 3.1. Basic Idea

The state of the quantum circuit is represented as nodes in a bundle search tree. A single search in the beam search framework consists of three main processes: selection, expansion, and decision. In the selection process, a certain number of nodes are chosen from the current layer based on the effect values of each node, with the quantity determined by the width of the bundle search. The expansion process applies all candidate SWAP operations to the selected nodes, generating an equal number of expansion nodes, and calculates the effect values of each SWAP operation using an approximate real effect value function. The selection and expansion processes are repeated until the set search depth is reached. The decision process compares the long-term effect values of the newly expanded leaf nodes, then backtracks to the set of nodes generated during the first expansion. Ultimately, a new node is selected as the root node for the next search, which is an ancestor of the chosen leaf nodes.

The following explanation combines with Figure 4 to provide a detailed illustration. The state of the quantum circuit $s_{0}$ represented as the root node of the search tree after removing the executable gates under the initial mapping, is treated as the current layer (the blue box in Figure 4). Since the current layer contains only one node, the root node is selected for expansion. The specific steps involve inserting all candidate SWAP gates into the circuit of the root node and removing the quantum gates that become executable due to the insertion of the SWAP gates. Assuming there are currently three candidate SWAP gates, three nodes—let us call them $s_{1}$, $s_{2}$, and $s_{3}$—are expanded, and the long-term effect values for each corresponding node are calculated. At this point, the layer containing these nodes is the current layer (the green box in Figure 4). If the search width is set to 2, the two nodes with the highest values (the gray nodes $s_{2}$ and $s_{3}$ in Figure 4) are selected for further expansion. This process is repeated until the set search depth is reached, which we assume to be 4 in this case. During this phase, the long-term effect values of the leaf nodes in the last layer (the red box in Figure 4) are compared, and node $s^{\prime }$ is chosen for backtracking. From the set of ancestor nodes generated during the first expansion (the green box in Figure 4), node $s_{2}$ is decided upon as the root node for the next search, as shown in the (b) in Figure 4. This search process will continue to repeat until all quantum gates in the circuit have been executed.

### 3.2. Preprocessing

The preprocessing of the quantum circuit mapping algorithm based on the beam search framework primarily includes the following operations:

**Calculate the physical distance of double quantum gates**: Given a quantum architecture diagram AG, let *D* represent the distance matrix recording the shortest path lengths between various physical qubits, where the distance between each pair of adjacent qubits is 1. For a two-qubit gate, suppose we have AG, π, the two-qubit gate $g(v_{i},v_{j})$, and the corresponding logical qubits are $v_{i}$ and $v_{j}$, where i≠j. In this case, the physical distance of the two-qubit gate is the distance between the physical qubits $q_{i}$ and $q_{j}$ minus one, denoted as $d_{gate}$, which can be expressed as follows:(1)$$d_{gate}=D\left[\pi \left(v_{i}\right)\right]\left[\pi \left(v_{j}\right)\right]-1$$
where $\pi (v_{i})$ represents the physical qubit corresponding to the logical qubit under the mapping π.

**Quantum gate dependency graph (directed acyclic graph, DAG)**: Any quantum circuit can be represented as a sequence of combinations of single quantum gates and CNOT gates. Therefore, in this context, the term ’double quantum gate’ specifically refers to the CNOT gate. Figure 5 illustrates a specific quantum gate dependency graph. To clearly demonstrate the dependency relationships of CNOT gates on physical qubits within the quantum circuit, a directed acyclic graph (DAG) is used to represent the dependency order of these CNOT gates. Since single quantum gates operate only on single qubits and do not create dependencies on other qubits, the circuit mapping algorithm does not consider the influence of single quantum gates in its design. A CNOT gate can only be executed when both corresponding physical qubits $q_{i}$ and $q_{j}$ have completed all preceding CNOT gates.

**Quantum circuit layering**: The layering of quantum circuits is achieved through the quantum gate dependency graph, and based on this layered structure, the definition of the front layer is proposed. The front layer is defined as a set of CNOT gates that have no predecessor dependencies; there are no unexecuted predecessor CNOT gates in the DAG, allowing these gates to be executed immediately on the hardware. Specifically, for a CNOT gate $\mathrm{CNOT}(q_{i},q_{j})$, it can be included in the front layer when its corresponding two qubits $q_{i}$ and $q_{j}$ and all previously executed quantum gates on them have completed execution. As shown in the front layer of Figure 5, the CNOT gates $g_{1}$ and $g_{2}$ in the front layer have no pending predecessor quantum gates in DAG; thus, they can be safely placed in the front layer.

**Initial mapping of qubits**: Due to the use of different initial mapping schemes, varying results may be generated for different quantum circuits, and the optimal initial mapping scheme applicable to all quantum circuits remains unclear. Therefore, to avoid interference from the initial mapping of qubits on the quantum circuit mapping experiment, this study adopts only the most basic initial mapping strategy. In this scheme, the specific conditions of the CNOT gates in the front layer F are not considered; logical qubit $v_{i}$ is mapped to physical qubit $q_{i}$(i≥1 and i≤n), *n* is the number of qubits. This initial mapping does not take into account the details of the actual quantum circuit and its physical architecture, thereby eliminating any potential impact on subsequent quantum circuit mapping algorithms.

### 3.3. Approximate Real Effect Value Function

The cost function is the core of heuristic search algorithms [3,14], providing a basis for action decisions by estimating the favorability of candidate operations for problem-solving. Currently, existing heuristic quantum circuit mapping algorithms [3,14] generally use a lookahead cost function, which approximates the potential improvement from applying a SWAP gate by calculating the sum of the physical distances of all quantum gates within the lookahead window, ultimately selecting the SWAP gate with the minimum cost function value as the next action. However, the lookahead cost function only considers the direct impact of the SWAP gate on the physical distances of subsequent quantum gates, without further assessing the potential negative effects that this SWAP gate might introduce. For example, if the currently inserted SWAP gate increases the physical distance of a certain quantum gate within the lookahead window, the execution of that quantum gate may be adversely affected. To address this issue, additional SWAP gates need to be introduced in the subsequent mapping process to offset the negative effects of the aforementioned SWAP gate. To improve the shortcomings of the existing lookahead cost function in current mapping methods, this approach constructs a cost function that comprehensively considers both the physical distances of quantum gates and the SWAP gates needed thereafter. To distinguish it from the existing lookahead cost function, it is referred to as the ‘approximate real effect value function’.

The calculation idea of the approximate real effect value function is as follows: when evaluating the overall effect of a SWAP gate, the first step is to calculate the reduction in the sum of the physical distances of all quantum gates within the lookahead window after applying the SWAP gate, which serves as the numerator of the effect value function. Secondly, the number of subsequent SWAP gates that may be introduced within the lookahead window after applying this SWAP gate is estimated, which serves as the denominator of the effect value function. Next, the implementation details of this effect value function will be explained in detail.

According to the previous definition of the physical distance for double quantum gates, when $d_{gate}>0$, it implies that at least a number of SWAP gates equal to $d_{gate}$ must be inserted to satisfy the connectivity constraints between two qubits in the physical quantum architecture. Based on this, the effect value $H_{g}$ of a double quantum gate *g* under the action of candidate SWAP gates can be defined as follows:(2)$$H_{g}=D\left[\pi \left(g.v_{i}\right)\right]\left[\pi \left(g.v_{j}\right)\right]-D\left[\pi_{1}\left(g.v_{i}\right)\right]\left[\pi_{1}\left(g.v_{j}\right)\right]$$
where $\pi_{1}$ is the temporary mapping generated after applying the SWAP operation to π. For all double quantum gates in the front layer F, the effect value of the front layer F under the action of candidate SWAP gates can be obtained by summation, as follows:(3)$$H_{F}=\sum_{g\in F}H_{g}$$

Since the range of change in physical distance for double quantum gates due to the SWAP operation does not exceed 1, the range of $H_{g}$ is −1,0,1. When $H_{g}$ is positive, it indicates that this SWAP operation helps to reduce the physical distance of the current double quantum gate, making its execution more favorable. For $H_{F}$, if its value is positive, it is considered that the SWAP operation can effectively reduce the overall physical distance of the front layer F, thereby facilitating the advancement of the front layer in the DAG. For the quantum circuit state s=(π,PC,LC), there may be multiple candidate SWAP gates that can be inserted, so each candidate SWAP operation needs to be evaluated. For those SWAP operations with a negative $H_{F}$, to avoid situations where the algorithm does not converge, such operations are not considered.

For candidate SWAP gates, if their effect value $H_{F}$ is positive, subsequent operations can continue. In this case, using the multi-layer window approach to calculate the approximate real effect value function, if a negative situation arises as the front layer advances, it indicates that the initial candidate SWAP gate has a negative effect on the subsequent layer, resulting in an increased physical distance for that layer. To eliminate this negative effect, it is necessary to insert auxiliary SWAP gates identical to the original SWAP gate before that subsequent layer. However, such an operation will also negate the positive effect of the original candidate SWAP gate on the subsequent layer. Therefore, for subsequent layers that are not affected by the negative effects of the original candidate SWAP gate, it is necessary to insert auxiliary SWAP gates identical to the original candidate SWAP gate before them to retain the positive effect of the original candidate SWAP gate.

The following is a detailed explanation in conjunction with Figure 6. In Figure 6, the original candidate SWAP gate is represented in red, and the multi-layer window size is set to 5, indicating that the front layer *F* needs to be advanced five times. First, Layer 1 is considered the front layer of the DAG. Since $H_{F}$ has been calculated and is positive, there is no need to insert auxiliary SWAP gates before it; the front layer *F* can be advanced directly to the next layer in the DAG, which is Layer 2. Next, the effect value $H_{F}$ for this layer is calculated. Assuming it is negative, an auxiliary SWAP gate identical to the original candidate SWAP gate must be inserted before this layer, reverting the new mapping $\pi^{\prime }$ back to the original mapping π, and recalculating $H_{F}$ for this layer, resulting in 0. Subsequently, the front layer *F* is advanced again, updating to Layer 3, and the effect value $H_{F}$ for this layer is calculated, assuming it is 0. Since an auxiliary SWAP gate was previously added to restore the mapping to π, eliminating the positive effect of the original candidate SWAP gate on this layer, it is necessary to add another auxiliary SWAP gate identical to the original SWAP gate before this layer to restore its positive value. These operations will continue to be repeated until the window advancement is complete.

After calculating the effect values for each layer of the window layer, the effect value of node *s* is obtained, expressed as follows:(4)$$VAL\left(s\right)=\frac{1}{\text{added\_swaps}}\sum_{f\in layer}\left(H_{f}\cdot decay\right)$$

Here, added_swaps represents the number of additional auxiliary SWAP gates added during the multi-layer window calculation process. For each layer in the window, a weight parameter is introduced to reduce the influence of subsequent layers, which will help the algorithm select candidate SWAP gates with parallel execution capabilities, thereby introducing a decay effect into the effect value function. The decay in Equation (Equation 4) refers to the decay parameter, which has been repeatedly verified through multiple experiments, with an adjustment to 0.7 yielding optimal results for the algorithm. Additionally, VAL(s) is used to quantify the effect value of the new node *s* after applying the SWAP operation. Clearly, as VAL(s) increases, the number of SWAP gates needed to reach the target state will correspondingly decrease, thereby enhancing the superiority of node *s*.

### 3.4. Beam Search Framework

The beam search method, as a combinatorial optimization algorithm, is an improvement over the greedy strategy. In the search process at each layer, this method retains multiple candidate results instead of just the current optimal output. It does not search the entire solution space but improves the algorithm’s computational speed by branching, selecting, and eliminating within the solution space. It is known that the quantum circuit mapping optimization problem is NP-hard, with a vast search space, making it extremely difficult to find the optimal solution. Moreover, as a critical part of the quantum computing compilation process, the execution time of the algorithm must be strictly controlled. Therefore, finding an approximately optimal solution has become a feasible approach.

Using the beam search method, with the goal of minimizing the number of inserted SWAP gates, we start from the initial state $s_{0}=\left(\pi_{ini},PC_{0},LC_{0}\right)$ to construct the beam search decision tree. Here, $\pi_{ini}$ represents the initial mapping, $PC_{0}$ is the physical quantum circuit applicable to the actual architecture, and $LC_{0}$ is the logical quantum circuit to be processed. The state s=(π,PC,LC) serves as a node in the tree, where PC represents the physical circuit of the current node, and LC represents the logical circuit of the current node. The effect value VAL(s) for each node is calculated using the approximate real effect value function. Through this process, the minimum number of SWAP gates required to reach the target node $s_{target}=\left(\pi ,PC,LC\right)$ (where LC is empty, indicating the target node) is computed.

Each node *s* in the beam search tree has three effect values for decision assessment: the reward value $REW(s,s^{\prime })$, the effect value VAL(s), and the long-term effect value $VAL_{long}\left(s\right)$. The reward value $REW(s,s^{\prime })$ indicates the extent to which the number of quantum gates to be executed in the circuit decreases from the parent node *s* to the child node $s^{\prime }$ after inserting a SWAP gate. The effect value VAL(s) describes the pros and cons of inserting a single SWAP gate in subsequent mapping tasks. The long-term effect value $VAL_{long}\left(s\right)$ reflects the overall quality of the current circuit state at each node (which includes all previously inserted SWAP gates) in the mapping task, serving as a basis for evaluation and decision-making during selection and backtracking. By applying a SWAP operation to node *s*, a new node is obtained, denoted as $s^{\prime }$, with *s* as its parent node. Thus, the reward value $REW(s,s^{\prime })$ from node *s* to $s^{\prime }$ can be defined, and its calculation function is as follows:(5)$$REW\left(s,s^{\prime }\right)=num\left(s\right)-num\left(s^{\prime }\right)$$

Here, num(s) represents the number of double quantum gates in the logical circuit of node *s*, so $REW(s,s^{\prime })$ can also refer to the number of newly added executable gates in the node $s^{\prime }$ compared to the node *s*, with its value always being non-negative. Based on the above definition of the nodes in the beam search tree, the beam search problem for quantum circuits can be divided into three processes: selection, expansion, and decision, as shown in Figure 7.

**Selection**: The selection process aims to find the optimal set of nodes in the beam search tree for further expansion. Part a of Figure 5 details the specific process of selection. For each layer of expanded nodes, a fixed number of nodes is selected as candidates for further expansion. First, the root node is added to the set of nodes to be selected, $selection=s_{0}$. Before reaching the search depth set by the beam search, a certain number of optimal nodes are selected from selection based on the long-term effect values of each node. The specific quantity *k* is determined by the search width width and the number of elements *n* in the set, specifically k=min(n,width). Through repeated experiments, it has been found that setting the search width of the beam search tree to 8 yields optimal results for the algorithm.

During the selection process, nodes are compared and ranked based on their long-term effect values, with the specific calculation function detailed in the expansion process. In each round of selection, the nodes with the highest long-term effect values are chosen to be added to the set of nodes for expansion, expansion.

**Expansion**: The expansion process involves applying SWAP operations to all nodes in the set of nodes for expansion, thereby generating the corresponding set of child nodes and calculating the long-term effect value for each node corresponding to the SWAP operation. The set of candidate SWAP operations for node *s* is denoted as SWAPs. For any SWAP gate $SWAP(v_{i},v_{j})$ in this set, the logical qubit $v_{i}$ corresponds to the physical qubit $\pi^{-1}\left(v_{i}\right)$, and $v_{j}$ corresponds to $\pi^{-1}\left(v_{j}\right)$, both of which should be part of the front layer *F* involved in node *s*. For a given node *s*, expansion can only be performed using the SWAP gates from its corresponding SWAPs set, a strategy that has been widely used in quantum circuit mapping.

By applying the candidate SWAP operations to s=(π,PC,LC), a new node $s^{\prime }=\left(\pi^{\prime },PC^{\prime },LC^{\prime }\right)$ is obtained, where $\pi^{\prime }$ is derived from π after the SWAP operation. Specifically, this involves remapping the two logical qubits of $SWAP(v_{i},v_{j})$ as $v_{i}\to \pi^{-1}\left(v_{j}\right)$ and $v_{j}\to \pi^{-1}\left(v_{i}\right)$. The set $LC^{\prime }$ is obtained by removing all executable gates in LC under the new mapping $\pi^{\prime }$, while $PC^{\prime }$ is formed by adding $SWAP(v_{i},v_{j})$ to PC and subsequently removing the executable gates from LC. Next, the long-term effect value of the new node $s^{\prime }$ is calculated, with the calculation function as follows:(6)$$VAL_{long}\left(s^{\prime }\right)=VAL_{long}\left(s\right)+REW\left(s,s^{\prime }\right)+VAL\left(s^{\prime }\right)$$

Here, *s* represents the parent node of s’. $VAL_{long}\left(s\right)$ is the long-term effect value of the parent node *s*; for the root node, which does not have a parent node, its long-term effect value is 0. Using the aforementioned approximate real effect value function, the one-step effect value $VAL(s^{\prime })$ for node $s^{\prime }$ can be calculated, while $REW(s,s^{\prime })$ represents the reward value obtained by applying the SWAP operation on the parent node *s* to derive the child node $s^{\prime }$. All newly expanded nodes are then added to the set of nodes for selection.

**Decision**: After repeatedly performing the selection and expansion processes for μ rounds, a decision is made, where μ is the search depth of the beam search tree. Through repeated experiments, it has been found that setting the search depth of the beam search tree to 13 yields optimal results for the algorithm. Based on the long-term effect values of each node in the current selection set, the leaf node with the highest long-term effect value is chosen, and the search backtracks to the initial set of nodes expanded from the root node to find the corresponding ancestor node. The SWAP operation corresponding to this ancestor node is selected as the final choice in the decision-making process, representing the long-term optimal SWAP operation for the root node state. This ancestor node is then set as the new root node, initiating the next search. When the LC corresponding to the new root node $s_{0}=\left(\pi ,PC,LC\right)$ is empty, it indicates that the quantum circuit mapping has been completed, and its PC is the desired quantum circuit that satisfies the physical quantum architecture AG.

### 3.5. Complexity Analysis of Proposed Algorithm

The time complexity of the approximate real effect value function is as follows: when the value of the current layer $H_{F}$ is 0, if added_swaps is not 0, it indicates that the previously inserted auxiliary SWAP gates have restored the mapping to the state before the original alternative SWAP gates were inserted, ensuring that the subsequent layer’s value $H_{F}$ is not negative. When encountering another layer with $H_{F}$ equal to 0, it can be inferred that it was originally positive but has become 0 due to the influence of the auxiliary SWAP gates. In this case, the same auxiliary SWAP gates need to be reinserted to restore it to a positive value. In terms of algorithm complexity, since the values of each layer can be fully computed after inserting the first SWAP gate, the time complexity of this effect value function is O(k), where *k* is the window size.

In the worst-case scenario, the algorithm’s time complexity is given by O(|E|·|G|·d·w·k), where |G| is the number of two-qubit gates in the quantum circuit, which indicates that in the worst case, inserting a SWAP gate can only execute one two-qubit gate. |E| represents the number of qubits in the quantum physical architecture. In the worst case, all SWAP gates applied to the qubits in the architecture diagram can serve as alternative SWAP gates, *d* denotes the depth of the clustered search tree, *w* is the width of the clustered search tree, and *k* represents the window size of the approximated real effect value function. Although the algorithm may be unsolvable in terms of time in the worst case, a balance can be achieved between algorithm performance and efficiency by adjusting *d*, *w*, and *k*. The optimal case is O(1), meaning the circuit already meets the constraints of the physical architecture and does not require mapping.

### 3.6. Practical Example

The following provides a practical example, where a simple quantum circuit is selected, as shown in the Figure 8a, which contains five CNOT gates and six single-qubit gates. It is assumed that this circuit will run on the physical architecture in Figure 3, with a search depth set to 15 and a search width set to 8. The initial mapping $\pi_{0}$ is defined as $v_{0}\to q0,v_{1}\to q_{1},\ldots ,v_{i}\to q_{i},\ldots$. First, as shown in Figure 8b, a clustered search tree is constructed. The root node $s_{0}$ contains the complete physical circuit PC, an empty logical circuit LC, and its corresponding mapping $\pi_{0}$. For the root node $s_{0}$, its front layer includes a $CNOT(q_{0},q_{2})$. Since this CNOT gate cannot be executed directly under the corresponding mapping at this node, an auxiliary SWAP is required to change its mapping, allowing the circuit to proceed. For this node, the available SWAP gate set is $SWAP\left(q_{0},q_{1}\right),SWAP\left(q_{1},q_{2}\right),SWAP\left(q_{2},q_{3}\right)$. Next, using the approximate real effect value function from Section 3.2, the effect value of the SWAP(q1,q2) gate is calculated with a window size set to 7. It can be observed that after inserting SWAP(q1,q2), the physical distances of the first CNOT(q0,q2) and the fourth CNOT(q0,q2) both decrease by 1, while the physical distance of the final CNOT(q0,q1) increases by 1, with no change in the physical distances of the other CNOT gates. Therefore, an additional SWAP gate needs to be added before the final CNOT(q0,q1). Given a decay parameter set to 0.7, the effect value for SWAP(q1,q2) is calculated as 1+1×0.7×0.7. These SWAP gates are applied to the circuit at the root node, resulting in the expansion of two new nodes. Each new node’s mapping is obtained by applying the corresponding SWAP gates to the mapping of the root node. The circuit is then advanced under the new mapping until no further execution is possible. At this point, combining the previously calculated effect values of the SWAP gates with the number of CNOT gates executed during the circuit advancement, the effect values of the expanded nodes are calculated to be 1.74 and 3.49, respectively. Since the number of expanded nodes is less than the search width, all expanded nodes are chosen as candidate nodes. As the search depth has not yet been reached, further expansion of the candidate nodes is performed, resulting in the expansion of four new nodes. Since node $s_{5}$ is obtained by adding another SWAP(q1,q2) to $s_{2}$(SWAP(q1,q2)), as shown in the Figure 8b, it has already executed all the gates in the circuit, thus completing the algorithm.

## 4. Results

### 4.1. Experimental Setup

The quantum circuit mapping algorithm based on the beam search framework aims to minimize the total number of inserted SWAP gates. Therefore, in the experiments, the number of CNOT gates inserted during the quantum circuit mapping process is used as an evaluation metric, which is three times the number of SWAP gates. The IBM Q Tokyo architecture is chosen as the quantum physical architecture for the experiments. This quantum computing platform is a commonly used testing platform in recent similar studies and contains 20 qubits, as detailed in Figure 2. The experiments utilize quantum circuits that are widely used as benchmarks in similar research; these benchmark circuits consist of the quantum instruction set supported by IBM Q series quantum computers, where double quantum gates only include CNOT gates, and the rest are single quantum gates. The algorithm is implemented in Python 3.10, and the experiments are conducted on a Lenovo Y9000P laptop equipped with an Intel Core i9-13900HX CPU (manufactured by Intel Corporation, Santa Clara, CA, USA) and 16GB of memory (Samsung Electronics, Suwon, Republic of Korea), running Windows 11 operating system.

To evaluate the performance of the algorithm, the SABRE algorithm proposed in reference [3] and the IBM basic mapping algorithm were chosen as comparison benchmarks. The SABRE algorithm is a commonly used comparative algorithm in quantum circuit mapping research, while the IBM basic mapping algorithm is the most widely used and fundamental mapping algorithm in current quantum computing platforms. After multiple tests, the quantum circuit mapping algorithm based on the beam search framework finalized a set of optimal parameters, specifically as follows: the search width of the beam search framework is set to 8, the search depth to 15, the decay parameter in the approximate real effect value function is set to 0.7, and the window size for multi-layer calculations of the effect value function is set to 7.

### 4.2. Comparison of Experimental Results

The quantum circuit mapping method based on beam search does not consider the optimization of results based on the initial mapping. Therefore, it initially uses the same identity mapping, mapping the *i*th logical qubit to the *i*th physical qubit. The experimental results of the algorithm on multiple benchmark circuits are shown in Table 1. The first two columns represent the names of the quantum circuits and the number of CNOT gates included, while the next three columns display the number of CNOT gates inserted by the SABRE algorithm, the IBM basic mapping algorithm, and our BSF algorithm, respectively. To intuitively show the difference between this algorithm and the existing algorithms, the following two columns present the reduction in the number of CNOT gates inserted by this algorithm compared to the other two methods. This value is calculated by subtracting the number of CNOT gates inserted by this algorithm from that of the other algorithms, and then dividing by the number of gates inserted by the other algorithms. The last column of Table 1 provides the runtime of this algorithm.

As shown in Table 1, the quantum circuit mapping algorithm based on the bundle search framework significantly reduces the number of additional CNOT gates inserted in medium-sized (original CNOT gate count over 1000) and large circuits (original CNOT gate count over 10,000) compared to the SABRE algorithm in reference [3], with an average reduction exceeding 44%. However, it is somewhat lacking in small circuits. In contrast to IBM’s basic mapping algorithm, this algorithm can significantly reduce the number of additional CNOT gates inserted in small, medium, and large circuits, achieving an average reduction of over 60%.

The disadvantage of this algorithm on small circuits may stem from the impact of the initial mapping on the results. To explore the enhancement effect of the initial mapping on algorithm performance, we adopted the initial mapping method from reference [3] and presented the experimental results of the three algorithms on multiple benchmark circuits, as shown in Table 2. The first column lists the names of the quantum circuits, while the next three columns show the number of CNOT gates inserted by the SABRE algorithm with the initial mapping, the algorithm without initial mapping (BSF), and the algorithm with initial mapping (BSF’), respectively. The following two columns display the reductions in the number of inserted CNOT gates of the algorithm compared to the other two methods after applying the initial mapping. The last column of Table 2 provides the average runtime of the algorithm.

As shown in Table 2, after applying the initial mapping, this algorithm can reduce the number of inserted CNOT gates by an additional 10% compared to the SABRE algorithm for medium-sized and large circuits, based on the no-initial-mapping scenario. Although the algorithm still struggles with some small circuits where it cannot effectively reduce the number of inserted CNOT gates, significant improvements have been observed in certain cases. By comparing the results of the algorithm before and after applying the initial mapping, it can be confirmed that the initial mapping positively impacts the algorithm’s performance, resulting in an average reduction of 16.86% in the number of inserted gates.

By applying the initial mapping, the discrepancies in results caused by the initial mapping between different algorithms are eliminated. Compared to this algorithm, the SABRE algorithm only considers the foresight in the initial mapping and cost function, lacking the global perspective that this algorithm employs during the search process. Therefore, in large test cases, as the logical circuit deepens, the influence of the initial mapping gradually diminishes, allowing this algorithm to insert fewer CNOT gates to generate the final circuit. To address the issue of excessive execution time for this algorithm on certain large circuits, adjustments can be made to the width and depth of the bundle search to achieve a balance between execution time and algorithm performance.

## 5. Conclusions

The quantum circuit mapping algorithm based on the bundle search framework proposed in this study achieves efficient mapping of logical quantum circuits, allowing them to overcome connectivity constraints of quantum physical architectures and successfully deploy on specific quantum computing devices. This method significantly reduces the number of basic gates required for quantum circuit mapping by introducing a multi-layer window approximation of the true effect value function within the bundle search framework. Experimental results show that for large circuits with over 10,000 CNOT gates, this method can quickly generate mapping schemes with fewer inserted SWAP gates, demonstrating generally superior performance compared to existing methods. However, it shows some shortcomings on certain small circuits due to the lack of consideration for the impact of initial mapping. Looking ahead, as the number of qubits in NISQ devices increases dramatically, the scale of quantum circuits will also become increasingly large. Therefore, this algorithm is of significant importance for enhancing the usability and computational efficiency of NISQ computing devices. Future work will focus on designing suitable initial mappings for this algorithm and improving the effect value function to address the algorithm’s shortcomings encountered during experimental processes.

## Figures and Tables

**Figure 1 entropy-27-00232-f001:**
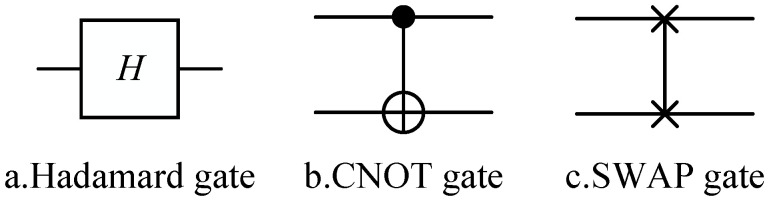
Hadamard, CNOT, SWAP.

**Figure 2 entropy-27-00232-f002:**
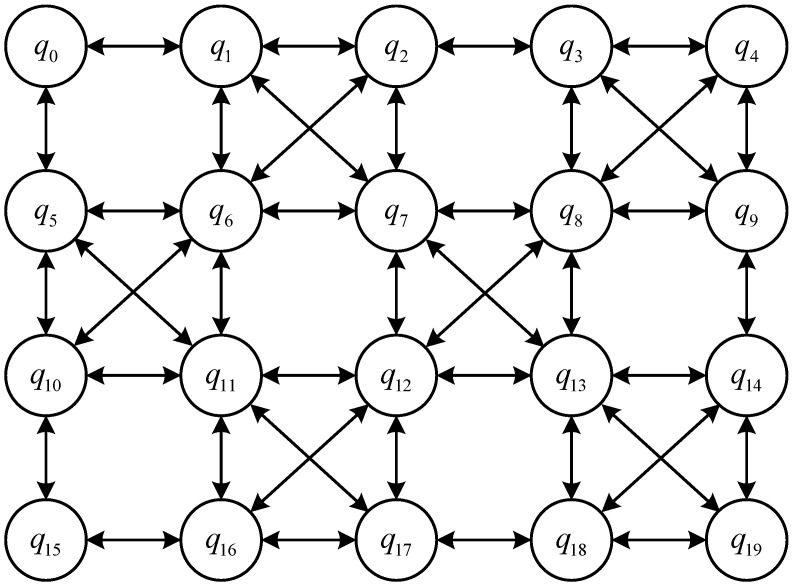
IBM Q Tokyo quantum physical architecture.

**Figure 3 entropy-27-00232-f003:**
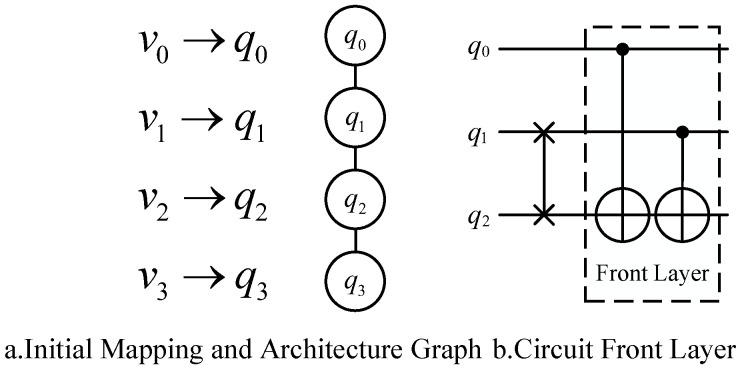
Initial mapping, physical architecture, and circuit front layer: a simple mapping example.

**Figure 4 entropy-27-00232-f004:**
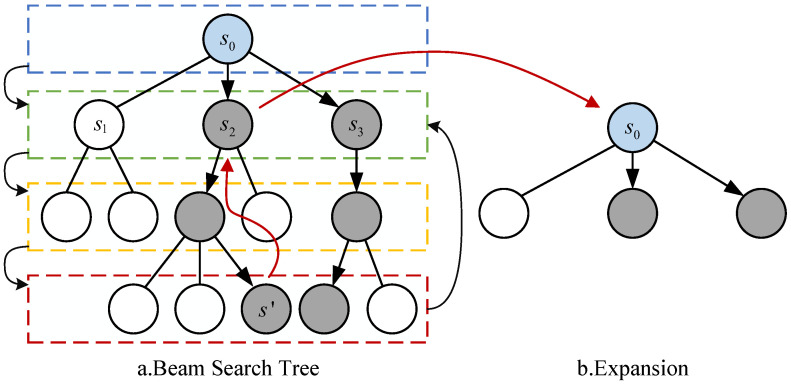
Schematic diagram of the beam search framework.

**Figure 5 entropy-27-00232-f005:**
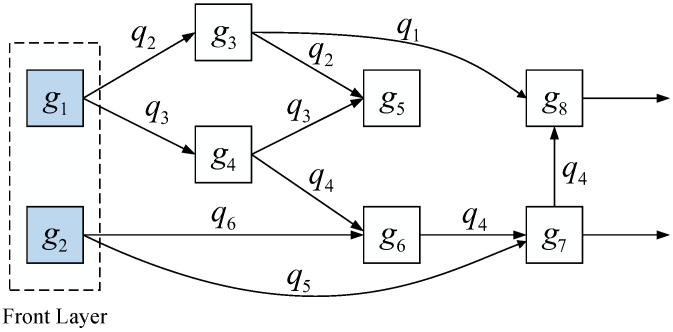
Quantum gate dependency graph represented by a directed acyclic graph (DAG).

**Figure 6 entropy-27-00232-f006:**
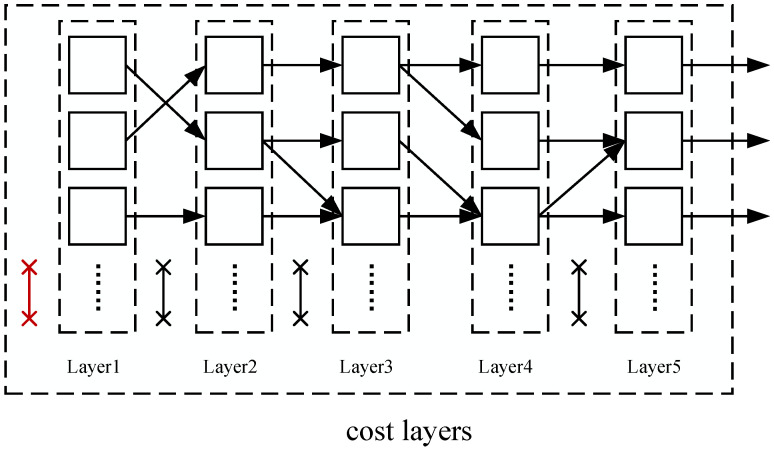
Schematic diagram of the approximate real effect value function.

**Figure 7 entropy-27-00232-f007:**
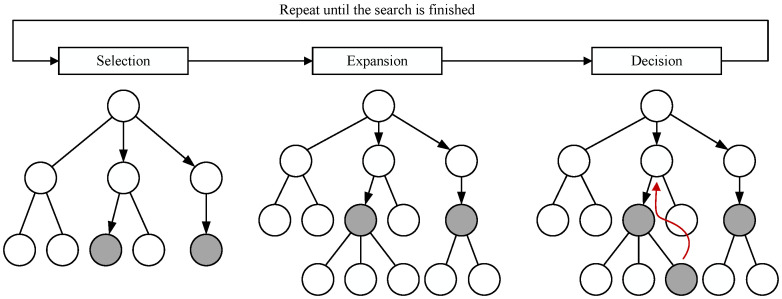
Schematic diagram of the beam search framework.

**Figure 8 entropy-27-00232-f008:**
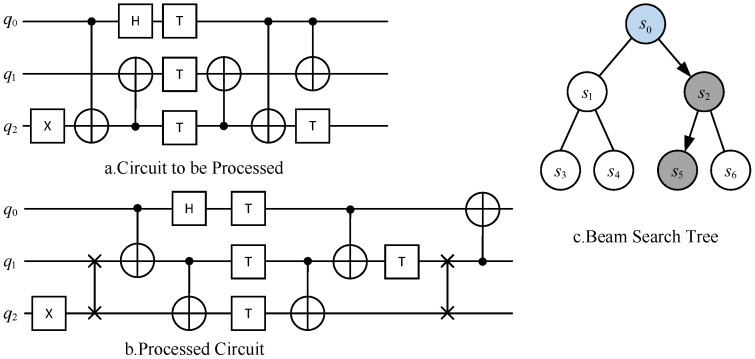
Practical example.

**Table 1 entropy-27-00232-t001:** IBM Q Tokyo comparison of experimental results.

					Comparison		
Circuit Name	CNOT Gates	SABRE	IBM Basic	BSF	With SABRE	With IBM Basic	Running Time/s
graycode6_47	5	0	9	9	-	0	0.01
qft_13	156	78	201	123	−57.69	38.81	9.33
qft_16	240	126	609	207	−64.29	66.01	23.56
rd84_142	154	87	285	123	−41.38	56.84	3.78
adr4_197	1498	1137	2070	777	31.66	62.46	24.58
radd_250	1405	1092	1914	681	37.64	64.42	14.35
z4_268	1343	1044	1803	579	44.54	67.89	15.57
sym6_145	1701	1086	1590	1002	7.73	36.98	7.39
misex1_241	2100	1131	1938	720	36.34	62.85	16.75
rd73_252	2319	1845	3249	1296	29.76	60.11	35.72
cycle10_2_110	2648	2097	3609	1074	48.78	70.24	31.56
square_root_7	3089	2442	6906	1263	48.28	81.71	121.79
sqn_258	4459	3651	5679	2154	41	62.07	66
rd84_253	5960	5772	7761	2916	49.48	62.43	154.36
root_255	7493	6246	9387	3351	46.35	64.3	168.61
co14_215	7840	7635	10,218	4161	45.5	59.28	256.9
mlp4_245	8232	7533	10,740	3945	47.63	63.27	214.99
urf2_277	10,066	9978	13,413	5697	42.9	57.53	433.68
clip_206	14,772	14,337	19,752	6549	54.32	66.84	845.89
sym9_193	15,232	16,002	22,068	6183	61.36	71.98	552.73
9symml_195	15,232	16,002	22,068	6183	61.36	71.98	561.66
hwb8_113	30,372	28,623	41,697	12,675	55.72	69.6	2671.63
sym10_262	28,084	28,779	39,882	11,748	59.18	70.54	2127.07

**Table 2 entropy-27-00232-t002:** Comparison of experimental results after applying initial mapping.

				Comparison		
Circuit Name	SABRE	BSF	BSF’	With SABRE	With BSF	Running Time/s
graycode6_47	0	9	0	-	100.00	0.00
qft_13	78	123	96	−23.08	21.95	6.28
qft_16	126	207	76	39.68	63.29	14.84
rd84_142	87	123	93	−6.90	24.39	2.86
adr4_197	1137	777	615	45.91	20.85	16.56
radd_250	1092	681	638	41.58	6.31	30.23
z4_268	1044	579	537	48.56	7.25	25.97
sym6_145	1086	1002	829	23.66	17.27	15.02
misex1_241	1131	720	483	57.29	32.92	26.77
rd73_252	1845	1296	1077	41.63	16.90	78.39
cycle10_2_110	2097	1074	1035	50.64	3.63	57.94
square_root_7	2442	1263	1104	54.79	12.59	151.33
sqn_258	3651	2154	1776	51.36	17.55	49.88
rd84_253	5772	2916	2796	51.56	4.12	127.59
root_255	6246	3351	3267	47.69	2.51	137.57
co14_215	7635	4161	3969	48.02	4.61	215.93
mlp4_245	7533	3945	3936	47.75	0.23	218.22
urf2_277	9978	5697	5295	46.93	7.06	548.78
clip_206	14,337	6549	6657	53.57	−1.65	885.76
sym9_193	16,002	6183	6372	60.18	−3.06	662.84
9symml_195	16,002	6183	6372	60.18	−3.06	657.97
hwb8_113	28,623	12,675	10,413	63.62	17.85	1587.14
sym10_262	28,779	11,748	10,074	65.00	14.25	1514.58

## Data Availability

Data are available upon request.

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
