# Peer review of "A Beam Search Framework for Quantum Circuit Mapping"

_entropy, 2025, doi:10.3390/e27030232_

Round 1

Reviewer 1 Report

Comments and Suggestions for Authors

Comments on the Quality of English Language

English Language quality can be improved.

Author Response

Comments 1:There are no edges with arrows in Figure2.

Response 1: Thank you for pointing this out. We agree with this comment. Therefore, we have modified Figure 2 by adding arrows to each edge.

,

Figure 2. IBM Q Tokyo Quantum Physical Architecture

Comments 2: Should the mapping  be please verify.

Response 2: Thank you for pointing this out. We agree with this comment. However, the description of this mapping is fine; it only needs to change the previous  and  to  and . Therefore, we have modified Section 2.4 as follows:

Figure 3. Initial mapping, physical architecture and Circuit Front Layer: a simple mapping example

        The basic quantum circuit mapping process is as follows: Assuming the quantum physical architecture is as shown in Figure 3, given an original quantum circuit diagram to be processed, as indicated in part b of the figure, only the  can be executed directly. The  cannot be executed immediately because its corresponding logical qubits,  and , do not correspond to directly adjacent physical qubits,  and , when mapped to the physical architecture. Therefore, a quantum circuit mapping transformation is required.

        Current mainstream mapping transformation methods involve inserting SWAP operations into the quantum circuit to exchange the states of the corresponding logical qubits in the physical qubits. This approach allows for adjustments in the mapping state, enabling the execution of two-qubit gates that are originally constrained by connectivity limitations. As shown in Figure 3, after inserting a SWAP gate between the logical qubits  and  of the second CNOT gate, all two-qubit gates in the updated front layer satisfy the connectivity constraints. After inserting the SWAP gate, the mapping  updates to , allowing all quantum gates in the front layer to be executable.

Comments 3: There are no nodes A, B, C and D in Figure4, please update the figure or update the text accordingly.

Response 3: Thank you for pointing this out. We agree with this comment. We update the text according to Figure 4, as follows:

Figure 4. Schematic Diagram of the Beam Search Framework

        The following explanation combines with Figure 4 to provide a detailed illustration. The state of the quantum circuit  represented as the root node of the search tree after removing the executable gates under the initial mapping, is treated as the current layer (the blue box in Figure 4). Since the current layer contains only one node, the root node is selected for expansion. The specific steps involve inserting all candidate SWAP gates into the circuit of the root node and removing the quantum gates that become executable due to the insertion of the SWAP gates. Assuming there are currently three candidate SWAP gates, three nodes—let's call them , , and  are expanded, and the long-term effect values for each corresponding node are calculated. At this point, the layer containing these nodes is the current layer (the green box in Figure 4). If the search width is set to 2, the two nodes with the highest values (the gray nodes  and  in Figure 4) are selected for further expansion. This process is repeated until the set search depth is reached, which we assume to be 4 in this case. During this phase, the long-term effect values of the leaf nodes in the last layer (the red box in Figure 4) are compared, and node  is chosen for backtracking. From the set of ancestor nodes generated during the first expansion (the green box in Figure 4), node  is decided upon as the root node for the next search, as shown in the (b) in Figure 4. This search process will continue to repeat until all quantum gates in the circuit have been executed.

Comments 4: The mapping  maps logical bit to a physical bit , if yes then equation 1 above should include  not

Response 4: Thank you for pointing this out. We agree with this comment. We have revised Equation 1 and Equation 2, as follows:

Comments 5: Provide a complexity analysis of the proposed heuristic search algorithms(best case and worst cases analysis)

Response 4: Thank you for pointing this out. We agree with this comment. We have provided an analysis of the algorithm's time complexity, as detailed in Section 3.5.

        The time complexity of the approximate real effect value function is as follows: when the value of the current layer  is 0, if  is not 0, it indicates that the previously inserted auxiliary SWAP gates have restored the mapping to the state before the original alternative SWAP gates were inserted, ensuring that the subsequent layer's value  is not negative. When encountering another layer with  equal to 0, it can be inferred that it was originally positive but has become 0 due to the influence of the auxiliary SWAP gates. In this case, the same auxiliary SWAP gates need to be reinserted to restore it to a positive value. In terms of algorithm complexity, since the values of each layer can be fully computed after inserting the first SWAP gate, the time complexity of this effect value function is , where  is the window size.

        In the worst-case scenario, the algorithm's time complexity is given by , where  is the number of two-qubit gates in the quantum circuit, which indicates that in the worst case, inserting a SWAP gate can only execute one two-qubit gate.  represents the number of qubits in the quantum physical architecture. In the worst case, all SWAP gates applied to the qubits in the architecture diagram can serve as alternative SWAP gates,  denotes the depth of the clustered search tree,  is the width of the clustered search tree, and  represents the window size of the approximated real effect value function. Although the algorithm may be unsolvable in terms of time in the worst case, a balance can be achieved between algorithm performance and efficiency by adjusting , , and . The optimal case is , meaning the circuit already meets the constraints of the physical architecture and does not require mapping.

Comments 5: The rest of the algorithm seems to be technically correct but is hard to follow, A detailed practical example is needed, please provide an example where the algorithm is applied on a simple quantum circuit such as the 4-bit parity generator, the full adder, the 4-bit ring counter or any other circuit you select. You need to show step by step and in detail the values you get when calculating equations 1 to 6and the step-by-step structure of the beam search tree.

Response 5: Thank you for pointing this out. We agree with this comment. We have provided a practical example, as detailed in Section 3.6.

Figure 8. Practical Example

        The following provides a practical example, where a simple quantum circuit is selected, as shown in the Figure 8.a, which contains 5 CNOT gates and 6 single-qubit gates. It is assumed that this circuit will run on the physical architecture in Figure 3, with a search depth set to 15 and a search width set to 8. The initial mapping  is defined as . First, as shown in Figure 8.b, a clustered search tree is constructed. The root node  contains the complete physical circuit , an empty logical circuit , and its corresponding mapping . For the root node , its front layer includes a . Since this CNOT gate cannot be executed directly under the corresponding mapping at this node, an auxiliary SWAP is required to change its mapping, allowing the circuit to proceed. For this node, the available SWAP gate set is . Next, using the approximate real effect value function from Section 3.2, the effect value of the  gate is calculated with a window size set to 7. It can be observed that after inserting $SWAP(q1, q2)$, the physical distances of the first  and the fourth  both decrease by 1, while the physical distance of the final  increases by 1, with no change in the physical distances of the other CNOT gates. Therefore, an additional SWAP gate needs to be added before the final . Given a decay parameter set to 0.7, the effect value for  is calculated as . These SWAP gates are applied to the circuit at the root node, resulting in the expansion of two new nodes. Each new node's mapping is obtained by applying the corresponding SWAP gates to the mapping of the root node. The circuit is then advanced under the new mapping until no further execution is possible. At this point, combining the previously calculated effect values of the SWAP gates with the number of CNOT gates executed during the circuit advancement, the effect values of the expanded nodes are calculated to be 1.74 and 3.49, respectively. Since the number of expanded nodes is less than the search width, all expanded nodes are chosen as candidate nodes. As the search depth has not yet been reached, further expansion of the candidate nodes is performed, resulting in the expansion of 4 new nodes. Since node  is obtained by adding another  to ( ), as shown in the Figure 8.b, it has already executed all the gates in the circuit, thus completing the algorithm.

Reviewer 2 Report

Comments and Suggestions for Authors

Dear authors,

Your manuscript titled “A Beam Search Framework for Quantum Circuit Mapping” is a well-written paper with interesting ideas solid results. The paper introduces a beam search framework for quantum circuit mapping on NISQ devices, addressing the NP-hard problem of optimizing qubit connectivity while minimizing SWAP gate insertions. Leveraging techniques inspired by natural language processing, it proposes an innovative approach to solving quantum circuit mapping problems. The work targets a critical challenge in quantum computing and presents meaningful experimental results, positioning itself as a valuable contribution to the field. However, before recommending it for publication, I do have a few comments I hope you can address.

1)      Line 80, what do you mean by effective time? Do you mean lifetime, or T1?

2)      Line 84, I do not see bidirectional arrows in Fig. 2. Please fix it.

3)      You lack captions throughout the entire paper. For Fig. 3, readers won’t understand what you are showing by just reading the caption, and you are missing labels of (a) and (b) based on reading the caption.

4)      Line 97, “the basic quantum … now introduced” doesn’t make sense at the beginning. Do you mean “introduced as follows?”

5)      Line 99 and 100: is not clear which CNOT is the first and which is the second.

6)      Line 108, again, I can’t see what is shown in Figure 3.

7)      Line 110, using the letter π to represent mapping should be introduced at an early place.

8)      Caption for Fig. 4 is similar to Fig. 3; it needs to be elaborated more.

9)      Line 121, you mentioned the cost function here. Can you give it some introduction?

10) Line 137, I don’t see B or C or D in Fig. 4. Please fix it.

11) Line 153: you have already defined g(vi,vj) in line 151, you don’t have to repeat it.

12) Line 159: CNOT should be introduced at a much earlier place, and don’t have to be introduced again here, and later places, for example line 170.

13) Line 160: do you mean Fig. 5 here?

14) Caption of Fig. 5 needs to be improved.

15) Line 176: what do you mean by “in-degree”?

16) Why is line 183 special in terms of defining the mapping? I mean why do you need to specifically say v1->q1 and then vi->qi?

17) Section 3.3 needs to have some more references. For example, what is cost function, how do you come to the conclusion of “currently, existing … ”? Many of the statements here need to have references.

18) You used the term “cost function” many times, I recommend use abbreviations.

19) In line 210 and line 212, you have mentioned “numerator” and “denominator”. At this point, you need to explicitly write down and define the cost function.

20) Line 217, you are missing “H” after “the effect value”.

21) Line 228, you need to define PC and LC.

22) I recommend using abbreviations when discussing about the Beam Search Framework as early as possible so you can make it easier for readers to follow and keep it consistent with Tab. 1.

23) Line 301, you don’t have to explain s’ and s again.

24) Line 335, why s’ is the parent node of s here?

25) Line 356-358. I am not sure if I can follow that using the number of CNOT instead of SWAP. Is there any problem to count the number of SWAP?

26) Table 1: the comparison should be in %, not just raw values, right? The first one also shouldn’t be 0/0. I get what you mean, but you can just cross it out, or write as -9/0.

27) Table 2: shouldn’t the bar be on top of “w/ SABRE”  and “w/ BSF”?

I think the paper will look much better if you can address the above comments, thank you!

Author Response

Comments 1: Line 80, what do you mean by effective time? Do you mean lifetime, or T1?

Response 1: Thank you for pointing this out. The effective time is the coherence time of a quantum gate, this refers to T1.

Comments 2: Line 84, I do not see bidirectional arrows in Fig. 2. Please fix it.

Response 2: Thank you for pointing this out. We agree with this comment. Therefore, we have modified Figure 2 by adding arrows to each edge.

Figure 2. IBM Q Tokyo Quantum Physical Architecture

Comments 3:  You lack captions throughout the entire paper. For Fig. 3, readers won’t understand what you are showing by just reading the caption, and you are missing labels of (a) and (b) based on reading the caption.

Response 3: Thank you for pointing this out. We agree with this comment. Therefore, we modified the caption.

Figure 3. Initial mapping, physical architecture and Circuit Front Layer: a simple mapping example

Comments 4: Line 97, “the basic quantum … now introduced” doesn’t make sense at the beginning. Do you mean “introduced as follows?”

Response 4: Thank you for pointing this out. We agree with this comment. Therefore, we have modified the text, as follows:

        The basic quantum circuit mapping process is as follows: Assuming the quantum physical architecture is as shown in Figure 3, given an original quantum circuit diagram to be processed, as indicated in part b of the figure, only the  can be executed directly. The  cannot be executed immediately because its corresponding logical qubits,  and , do not correspond to directly adjacent physical qubits,  and , when mapped to the physical architecture. Therefore, a quantum circuit mapping transformation is required.

Comments 5: Line 99 and 100: is not clear which CNOT is the first and which is the second.

Response 5: Thank you for pointing this out. We agree with this comment. Therefore, We specified the CNOT gate directly here, as follows:

        The basic quantum circuit mapping process is as follows: Assuming the quantum physical architecture is as shown in Figure 3, given an original quantum circuit diagram to be processed, as indicated in part b of the figure, only the  can be executed directly. The  cannot be executed immediately because its corresponding logical qubits,  and , do not correspond to directly adjacent physical qubits,  and , when mapped to the physical architecture. Therefore, a quantum circuit mapping transformation is required.

Comments 6: Line 108, again, I can’t see what is shown in Figure 3.

Response 6: Thank you for pointing this out. We agree with this comment. Therefore, we have modified Figure 3 by adding a SWAP gate between logical qubits  and , as shown in Figure 3.

Figure 3. Initial mapping, physical architecture and Circuit Front Layer: a simple mapping example

Comments 7:  Line 110, using the letter π to represent mapping should be introduced at an early place.

Response 7: Thank you for pointing this out. We agree with this comment. Therefore, we added this letter in the introduction section.

        Quantum circuit mapping consists of two parts: the initial mapping of qubits and the remapping. Since the initial mapping  can interfere with the remapping, the latter becomes the primary focus.

Comments 8: Caption for Fig. 4 is similar to Fig. 3; it needs to be elaborated more.

Response 8: Thank you for pointing this out. We agree with this comment. Therefore, we have modified the caption of Figure 4, as follows:

Figure 4. Schematic Diagram of the Beam Search Framework

Comments 9: Line 121, you mentioned the cost function here. Can you give it some introduction?

Response 9: Thank you for pointing this out. For an introduction to the cost function, you can refer to the beginning of Section 3.3, where it is discussed in detail.

Comments 10: Line 137, I don’t see B or C or D in Fig. 4. Please fix it.

Response 10: Thank you for pointing this out. We agree with this comment. We update the text according to Figure 4, as follows:

        The following explanation combines with Figure 4 to provide a detailed illustration. The state of the quantum circuit  represented as the root node of the search tree after removing the executable gates under the initial mapping, is treated as the current layer (the blue box in Figure 4). Since the current layer contains only one node, the root node is selected for expansion. The specific steps involve inserting all candidate SWAP gates into the circuit of the root node and removing the quantum gates that become executable due to the insertion of the SWAP gates. Assuming there are currently three candidate SWAP gates, three nodes—let's call them , , and  are expanded, and the long-term effect values for each corresponding node are calculated. At this point, the layer containing these nodes is the current layer (the green box in Figure 4). If the search width is set to 2, the two nodes with the highest values (the gray nodes  and  in Figure 4) are selected for further expansion. This process is repeated until the set search depth is reached, which we assume to be 4 in this case. During this phase, the long-term effect values of the leaf nodes in the last layer (the red box in Figure 4) are compared, and node  is chosen for backtracking. From the set of ancestor nodes generated during the first expansion (the green box in Figure 4), node  is decided upon as the root node for the next search, as shown in the (b) in Figure 4. This search process will continue to repeat until all quantum gates in the circuit have been executed.

Comments 11: Line 153: you have already defined g(vi,vj) in line 151, you don’t have to repeat it.

Response 11: Thank you for pointing this out. We agree with this comment. Therefore, we have modified text as follows:

        In this case, the physical distance of the two-qubit gate is the distance between the physical qubits  and  minus one, denoted as .

Comments 12: Line 159: CNOT should be introduced at a much earlier place, and don’t have to be introduced again here, and later places, for example line 170.

Response 12: Thank you for pointing this out. We agree with this comment. The introduction to the CNOT gate has already been covered in the previous 72 lines, and we removed the introduction here.

Comments 13: Line 160: do you mean Fig. 5 here?

Response 13: Thank you for pointing this out. We agree with this comment. It should refer to Fig.5 here.

Comments 14: Caption of Fig. 5 needs to be improved.

Response 14: Thank you for pointing this out. We agree with this comment. Therefore, we have modified the caption of Figure 5 as follows:

Figure 5. Quantum gate dependency graph, is represented by a Directed Acyclic Graph(DAG).

Comments 15: Line 176: what do you mean by “in-degree”?

Response 15: In computer graph theory, "in-degree" refers to the number of edges directed towards a specific node, indicating how many other nodes point directly to that node. To avoid causing misunderstandings for the reader, we describe the relationships between nodes directly, as follows:

        As shown in the Front Layer of Figure 5, the CNOT gates  and  in the front layer have no pending predecessor quantum gates in DAG, thus they can be safely placed in the front layer.

Comments 16: Why is line 183 special in terms of defining the mapping? I mean why do you need to specifically say v1->q1 and then vi->qi?

Response 16: Thank you for pointing this out. We agree with this comment. Therefore, we have revised the description here as follows:

        logical qubit  is mapped to physical qubit (  and ),  is the number of qubits.

Comments 17: Section 3.3 needs to have some more references. For example, what is cost function, how do you come to the conclusion of “currently, existing … ”? Many of the statements here need to have references.

Response 17: Thank you for pointing this out. We agree with this comment. Therefore, we added relevant citations at the beginning of Section 3.3, as follows:

        The cost function is the core of heuristic search algorithms [ 3, 15], providing a basis for action decisions by estimating the favorability of candidate operations for problem-solving. Currently, existing heuristic quantum circuit mapping algorithms [3 ,15 ] generally use a 190

lookahead cost function, which approximates the potential improvement from applying a SWAP gate by calculating the sum of the physical distances of all quantum gates within the lookahead window, ultimately selecting the SWAP gate with the minimum cost function 193

value as the next action.

Comments 18: You used the term “cost function” many times, I recommend use abbreviations.

Response 18: Thank you for pointing this out. This term is commonly used in computer science and is generally not abbreviated; therefore, we believe that using the term directly is appropriate.

Comments 19: In line 210 and line 212, you have mentioned “numerator” and “denominator”. At this point, you need to explicitly write down and define the cost function.

Response 19: Thank you for pointing this out. We agree with this comment. To distinguish it from the cost function used in other papers, we renamed our cost function as "Approximate Real Effect Value Function," with its specific definition shown in Formula 4.

Comments 20: Line 217, you are missing “H” after “the effect value”.

Response 20: Thank you for pointing this out. We agree with this comment. Therefore, we added  after "effect value."

Comments 21: Line 228, you need to define PC and LC.

Response 21: Thank you for pointing this out. We agree with this comment. Therefore, we added definitions for  and  in lines 284 and 285, as follows:

        The state  serves as a node in the tree, where  represents the physical circuit of the current node, and  represents the logical circuit of the current node.

Comments 22: I recommend using abbreviations when discussing about the Beam Search Framework as early as possible so you can make it easier for readers to follow and keep it consistent with Tab. 1.

Response 22: Thank you for pointing this out. We agree with this comment. Therefore, we used the abbreviation "BSF" in the abstract and introduction sections.

Comments 23: Line 301, you don’t have to explain s’ and s again.

Response 23: Thank you for pointing this out. We agree with this comment. Therefore, we removed the explanations for  and .

Comments 24: Line 335, why s’ is the parent node of s here?

Response 24: Thank you for pointing this out. We agree with this comment. Here, we wrote them incorrectly, it should be that  is the parent node of .

Comments 25: Line 356-358. I am not sure if I can follow that using the number of CNOT instead of SWAP. Is there any problem to count the number of SWAP?

Response 25: Thank you for pointing this out. In the context of quantum computing with superconducting devices, a SWAP gate is equivalent to the superposition of three CNOT gates. Since similar experiments typically use the number of CNOT gates as an evaluation metric, we use the number of CNOT gates here.

Comments 26: Table 1: the comparison should be in %, not just raw values, right? The first one also shouldn’t be 0/0. I get what you mean, but you can just cross it out, or write as -9/0.

Response 26: Thank you for pointing this out. We agree with this comment. Therefore, we made modifications to the contents of the table.

Comments 27: Table 2: shouldn’t the bar be on top of “w/ SABRE”  and “w/ BSF”?

Response 27: Thank you for pointing this out. We agree with this comment. Therefore, we made modifications to the contents of the table.
